# BST2, a Novel Inhibitory Receptor, Is Involved in NK Cell Cytotoxicity through Its Cytoplasmic Tail Domain

**DOI:** 10.3390/ijms231911395

**Published:** 2022-09-27

**Authors:** Jinsoo Oh, Eunbi Yi, Soo Kyung Jeong, Sehoon Park, Se-Ho Park

**Affiliations:** College of Life Science and Biotechnology, Korea University, 145 Anam-ro, Seongbuk-gu, Seoul 02841, Korea

**Keywords:** BST2, ITIM-motif, inhibitory receptor, natural killer cell, galectin-8, galectin-9

## Abstract

Bone Marrow Stromal Cell Antigen 2 (BST2) is a type II transmembrane protein expressed on various cell types that tethers the release of viruses. Natural killer (NK) cells express low levels of BST2 under normal conditions but exhibit increased expression of BST2 upon activation. In this study, we show for the first time that murine BST2 can control the cytotoxicity of NK cells. The cytoplasmic tail of murine BST2 contains an immunoreceptor tyrosine-based inhibitory motif (ITIM). The absence of BST2 on NK cells can enhance their cytotoxicity against tumor cells compared to wild type NK cells. NK cells isolated from NZW mice, which express ITIM-deficient BST2, also showed higher cytotoxicity than wild type NK cells. In addition, we found that galectin-8 and galectin-9 were ligands of BST2, since blocking galectin-8 or -9 with monoclonal antibodies enhanced the cytotoxicity of NK cells. These results suggested that BST2 might be a novel NK cell inhibitory receptor as it was involved in regulating NK cell cytotoxicity through its interaction with galectins.

## 1. Introduction

Bone Marrow Stromal Cell Antigen 2 (BST2; also known as tetherin, CD317, PDCA-1, and HM1.24) restricts the release of viruses at the plasma membrane. It consists of an N-terminal cytoplasmic tail, a transmembrane region, an extracellular coiled coil domain, and a C-terminal glycosylphosphatidylinositol (GPI) anchored domain [1,2]. BST2 can be translated into a long isoform or alternatively a short isoform, since the Kozak sequence flanking the first ATG (Methionine, M1) is leaky, and the downstream methionine (M13) appears to be an alternative translation start site [3]. Generally, the long isoform and the short isoform of BST2 are alternatively translated at an 1:1 ratio in a variety of cancer cell lines as well as in human CD4 T cells [3]. NZW mouse as a specific murine strain only expresses the short isoform of BST2 because the first ATG codon of *Bst2* gene is mutated, while the second one is available [4].

The N-terminal cytoplasmic tail region of BST2 has been reported to contain a putative hemi-immunoreceptor tyrosine-based activation motif (HemITAM) consisting of a single SH2-binding domain with a YXX-hydrophobic residue (YCRV in human, YLPV in mice) [5]. In HIV-1 infection, the ecto-domain of BST2 can restrict virus budding at the plasma membrane, while the hemITAM motif in the cytoplasmic tail of BST2 phosphorylated by Src-family kinases can activate NFκB signaling [5]. Thus, mice with cells expressing the short isoform of BST2 or NZW, which lose the hemiITAM motif, show reduced NFκB signaling with normal virus tethering abilities [4].

In this study, we found that the N-terminal cytoplasmic tail of murine BST2 also contained an immunoreceptor tyrosine-based inhibitory motif (ITIM, SFYHYL). ITIM-motif is a conserved sequence of amino acids (S/I/V/L) xYxx (I/V/L) in cytoplasmic tails of many inhibitory receptors [6]. When these receptors interact with specific ligands, the ITIM motif phosphorylated by Src kinase family proteins can recruit phosphotyrosine phosphatases (SHP-1 and SHP-2) or inositol-phosphatase SHIP [7]. These phosphatases can inhibit activating signals [8].

NK cells have activating and inhibitory receptors that can recognize a variety of ligands. Thus, NK cells play a key role in tumor surveillance and host defenses against viral infections [9]. Inhibitory receptors such as KIRs and CD94/NKG2A heterodimer on NK cells can recognize MHC class I molecules and then inhibit the killing of heathy cells, while NK cells can eliminate tumor cells or infected cells that do not express MHC class I according to the ‘missing self’ hypothesis [10,11]. Activating receptors, such as NKG2D or natural cytotoxicity receptors (NCRs), can recognize a variety of ligands such as MICA for NK2GD [12] and hemagglutinin (HA) of influenza virus for NCR1 [13]. NK cells can then directly eliminate target cells by secreting granules such as granzymes and perforin or by death receptor-mediated apoptosis [14]. They can also secrete various cytokines and chemokines including IFN-γ, MIP1-α, thereby enhancing anti-tumor responses [15]. Thus, NK cell activation and function are regulated by an interplay between inhibitory and activating receptors with a variety of ligands [9,16,17,18]. Cytotoxic T lymphocytes (CTLs) are also known to have an activation-dependent expression of immune checkpoint molecules [19]. Numerous studies have been conducted throughout the years in hopes of increasing the cytotoxicity (level) by blocking inhibitory receptors such as PD-1 and TIGIT on CTLs. Despite these efforts, therapy using CTLs is still incomplete. Thus, discovering new therapy targeting immune checkpoint molecules is being continued.

Galectins are carbohydrate-binding proteins with one or two carbohydrate-recognition domains (CRDs) that can bind to beta-galactoside-containing glycans. They are involved in multiple functions both intracellularly and extracellularly [20]. Galectins have been found in the cytosol, nucleus, extracellular matrix, and in circulation, thereby mediating intracellular signaling and cell adhesion [21]. Galectin-3 (Gal-3) can form oligomer to increase their binding avidity. Galectin-9 (Gal-9) is a tandem repeat-type connected by a flexible linker [20]. In cancer, Gal-3 and Gal-9 can suppress NK cytotoxicity by hindering NK receptor-ligand interaction. Thus, it has been suggested that galectins can regulate NK cell activity [22,23]. Galectin-8 (Gal-8), another tandem-repeat type of galectins, has been suggested to be associated with tumor cell adhesion, tumor cell survival, and metastasis [24]. Despite upregulation of Gal-8 during tumor progression in several cancers including lung cancer and head and neck cancer [25], the possibility of NK cell surveillance with Gal-8 is uncovered yet.

In this study, we found that BST2 of NK cells interacted with Gal-8 and Gal-9 and inhibited NK cell cytotoxicity. The inhibitory effect of Gal-8 and Gal-9 on NK cells was relayed through the ITIM motif on the cytoplasmic tail of BST2. Our findings suggest that BST2 might be a novel inhibitory receptor with an ITIM motif in the cytoplasmic tail that is involved in the suppression of NK cell activity by interacting with Gal-8 and Gal-9.

## 2. Results

This section may be divided by subheadings. It should provide a concise and precise description of the experimental results, their interpretation, as well as the experimental conclusions that can be drawn.

### 2.1. The Cytoplasmic Tail of BST2 Has an ITIM Motif

Through sequence comparison, we identified an ITIM motif, SxYxxL, on the cytoplasmic tail domain of murine BST2 and an ITIM-like motif, SxxYxxV, on the cytoplasmic tail of human BST2 (Table 1). Since the I/V/L/SxYxxL/V motif is a known as the ITIM motif [26,27], we hypothesize that BST2 might deliver inhibitory signals to cells which express BST2. The main role of an inhibitory receptor as a regulator is to control immune cells to not work excessively. If a population barely expresses BST2 at its naïve state, but highly induced when activated, it could mean that BST2 might be a regulator of that population. When splenocytes of naïve mice were analyzed by flow cytometry, myeloid cells such as macrophages and dendritic cells, especially plasmacytoid DC, expressed BST2 while lymphoid cells such as NK cells, T cells, and B cells hardly expressed BST2 (Appendix A). Expression of BST2 of lymphoid cells was increased after in vitro IL2 treatment (Appendix A). NK cells are part of an innate immunity. They are not restricted to a specific antigen. Thus, we first focused on NK cells. NK cells could also be activated by polyI:C. In comparison of NK cell ratio of *Bst2^+/+^* and *Bst2^-/-^* mice, there was no noticeable difference in splenic NK cell ratio upon polyI:C treatment (Appendix A). Hence, we sorted NK cells from polyI:C injected *Bst2^+/+^* and *Bst2^-/-^* mice and analyzed their cytotoxicity against target cells. The expression of BST2 on NK cells was upregulated after polyI:C stimulation (Figure 1A) and the cytotoxicity of *Bst2^+/+^* NK cells was significantly lower than that of *Bst2^-/-^* NK cells (Figure 1B).

To prepare killer cells with another method, we cultured splenocytes in vitro with IL-2 as described in Materials and Methods. In line with the results of the polyI:C injection, IL-2 stimulated NK cells showed an increased expression of BST2 (Figure 2A and Appendix A). *Bst2^+/+^* NK cells also showed a decreased cytotoxicity against target cells compared to *Bst2^-/-^* NK cells (Figure 2B and Appendix A). NK cells usually kill targets by secreting granules, such as granzyme B [30]. The amount of granzyme B in co-cultured medium of NK and target cells was also correlated with cytotoxicity. *Bst2^+/+^* NK cells secreted granzyme B less than *Bst2^-/-^* NK cells (Figure 2C). CD107a, a degranulation marker, was increased after the NK cells encountered their target cells [31]. In our analysis, the level of CD107a in *Bst2^+/+^* NK cells was lower than that in *Bst2^-/-^* NK cells, which were in direct proportion to their cytotoxicity (Figure 2D). To determine if *Bst2* gene knockout affected the status of NK cells and thus increased their cytotoxicity, expression levels of NKp46, NKG2A/C/E, NKG2D, 2B4, KLRG1, and CD25 were analyzed. We also compared the expression level of transcription factor T-bet. There were no significant differences in the expression of all the markers we tested (Appendix A). We also examined the intracellular expression of interferon gamma (IFN-γ) in *Bst2^+/+^* and *Bst2^-/-^* NK cells. Although IFN-γ expression increased after stimulation, both showed comparable levels of IFN-γ in the naïve state and after polyI:C stimulation (Appendix A).

To further test whether BST2 could act as an inhibitory receptor, we blocked BST2 on NK cells with a monoclonal antibody anti-BST2. Treatment with anti-BST2 antibody increased cytotoxicity of *Bst2^+/+^* NK cells (Figure 2E).

### 2.2. BST2 Can Bind to Galectin-8 and Galectin-9

Since previous results strongly suggested that BST2 might be a novel molecule related to NK cell cytotoxicity, we performed a pull-down assay to determine ligands of BST2 using human BST2 protein as a bait against human 293T cell lysate. Among several candidates that could be pulled-down, galectins caught our attention since Gal-3 and Gal-9 are known to dampen the activity of cytotoxic cells through interaction with NKp30 and TIM3, respectively [22,23]. To confirm that BST2 could bind to galectins specifically, we performed a reverse direction pull-down assay with alternative forms of GST-fused Gal-9 and Gal-8 molecules as baits against purified BST2 protein. BST2 was found to interact specifically with Gal-9 and Gal-8, but not with Gal-3 (Figure 3A,B). The N-terminal carbohydrate recognition domain (NCRD) of Gal-9 was a main part where BST2 bound (Figure 3B). BST2 also showed specific interaction with the NCRD of Gal-8 (Appendix A).

We examined whether Gal-9 was involved in NK cell cytotoxicity by performing NK cell cytotoxicity assay against B16 target cells in the presence of an anti-Gal-9 monoclonal antibody. For comparison, an isotype control antibody was used for treatment. Blocking Gal-9 substantially increased the cytotoxicity of *Bst2^+/+^* NK cells (Figure 3C, lane 1 vs. lane 2). However, the same treatment showed only a slight increase of cytotoxicity of *Bst2^-/-^* NK cells (lane 3 vs. lane 4), although they did have a higher basal level of cytotoxicity than *Bst2^+/+^* NK cells (lane 1 vs. lane 3). These slight increases in cytotoxicity might be due to the failure of Gal-9 interaction with other receptor(s) other than BST2. For example, TIM3 might be the receptor because it has been known to down-regulate LPS-triggered inflammatory responses by the interaction with Gal-9 [32]. On the other hand, others also reported that these inhibitory signals were relayed independent of TIM3 [22], which increases the possibility of BST2 participation in Gal-9 signaling. It suggests that TIM3 and BST2 are not single inhibitory factors for Gal-9 interaction, but factors that act in a cumulative way. Blocking Gal-8 also increased the cytotoxicity of *Bst2^+/+^* NK cells, while the cytotoxicity of *Bst2^-/-^* NK cells was not changed (Appendix A).

To confirm the interaction between Gal-9 and BST2, we analyzed cytotoxic effects of *Bst2^+/+^* and *Bst2^-/-^* NK cells against Gal-9-dificient (*Lgals9^-/-^*) YAC-1 cells. If *Lgals9^-/-^* target cells are unable to provide NK cells an inhibitory Gal-9 signal through BST2, lysis of *Lgals9^-/-^* target cells by *Bst2^+/+^* or *Bst2^-/-^* NK would be similar. While wild type YAC-1 cells were more potently killed by *Bst2^-/-^* NK cells than by *Bst2^+/+^* NK cells, cytotoxicities against *Lgals9^-/-^* YAC-1 cells by *Bst2^+/+^* NK cells and *Bst2^-/-^* NK cells were similar (Figure 3D). All three wildtype target tumor cell lines used in this study (RMA-S, B16 and YAC-1) showed an expression of Gal-9 (Figure 3E, upper panel), and *Lgals9^-/-^* YAC-1 showed complete loss of Gal-9 expression upon evaluation by western blotting (Figure 3E, lower panel). These results indicated that Gal-9 could deliver inhibitory signal by interacting with BST2 in addition to TIM3, thereby down-regulating NK cell cytotoxicity.

### 2.3. Long Isoform of BST2 Critically Downregulates the Cytotoxicity of NK Cells

We used NZW mice only expressing BST2 of short isoform (*Bst2^S/S^*), which lacked 12-amino acids including the ITIM motif at their N-terminus [4,33] to evaluate whether the inhibitory effect of BST2 in NK cells was delivered through the ITIM motif of its cytoplasmic tail. B6 *Bst2^+/-^* mice were mated with NZW mice. F1 hybrids having the same genetic background were used to generate NK cells (Figure 4A). The expression level of BST2 in NK cells was analyzed by flow cytometry staining. *Bst2^S/+^* NK cells were found to have a higher level of BST2 expression than *Bst2^S/-^* NK cells when they were stimulated with IL-2 as *Bst2^S/+^* NK cells had two copies of the *Bst2* gene (Figure 4B). We compared cytotoxicities of *Bst2^S/+^* NK cells and *Bst2^S/-^* NK cells against YAC-1, RMA-S, and B16 tumor cell lines. *Bst2^S/-^* NK cells were found to have higher cytotoxicity to all target tumor cells tested than *Bst2^S/+^* NK cells (Figure 4C). This suggests that the expression of the N-terminal cytoplasmic tail region containing ITIM motif in BST2 might be associated with the inhibition of NK cell cytotoxicity.

To verify whether the inhibitory signal was transduced through BST2, we compared cytotoxicities of *Bst2^S/+^* NK cells and *Bst2^S/-^* NK cells after treatment with anti-BST2 antibody. While blocking BST2 on *Bst2^S/+^* NK cells significantly enhanced NK cell cytotoxicity, the same blocking on *Bst2^S/-^* NK cells showed only a marginally insignificant enhancement of cytotoxicity (Figure 4D). This suggests that the N-terminal cytoplasmic tail of BST2 is critical to NK cell suppression. 

### 2.4. Cytoplasmic Tail ITIM Motif in BST2 Is Involved in NK Cell Cytotoxicity

Results shown in Figure 4 could be interpreted as a gene dosage effect since *Bst2^S/+^* NK cells have an extra copy of *Bst2* gene compared to *Bst2^S/-^* NK cells. To rule out gene dosage effects, we produced mice containing only a single copy of either long or short N-tailed-*Bst2* gene by backcrossing NZW mice at least eight times with B6 *Bst2^-/-^* mice. Resulting *Bst2^S/-^* mice were then intercrossed with *Bst2^+/-^* mice (Figure 5A). There were no noticeable differences in developmental stages of NK cells (immature NK (CD27^+^ CD11b^-^), transitional NK (CD27^+^CD11b^+^), and mature NK (CD27^-^ CD11b^+^) cells) among those mice (Appendix A). In addition, we analyzed the expression pattern of Ly49G2, an NK cell inhibitory receptor. However, we found no significant difference (Appendix A). Interestingly, *Bst2^S/-^* NK cells showed a higher BST2 expression level than *Bst2^+/-^* NK cells, although both mice had a single copy of *Bst2* gene (Figure 5B). The expression level of BST2 on *Bst2^S/-^* NK cells was even higher than that on *Bst2^+/+^* NK cells having two copies of the gene (Figure 5D). This might be attributable to the reduced endocytic recycling because of the lack of a full-size cytoplasmic tail [4].

Next, we investigated whether the long isoform rather than short isoform of BST2 served as an inhibitory receptor in NK cells. Cytotoxicities of *Bst2^+/-^* NK cells, *Bst2^S/-^* NK cells, and *Bst2^-/-^* NK cells from littermate control mice against YAC-1, RMA-S, and B16 cell lines were measured. *Bst2^+/-^* NK cells showed lower cytotoxicities than *Bst2^S/-^* NK cells and *Bst2^-/-^* NK cells, while *Bst2^S/-^* NK cells (pink columns) and *Bst2^-/-^* NK cells (gray columns) showed similar cytotoxicities to all cell lines tested (Figure 5C). In addition, we analyzed whether *Bst2^S/-^* NK cells could not show enhanced cytotoxicity against *Lgals9^-/-^* target cells as cases of anti-BST2 and anti-Gal-9 antibody treatments. While *Bst2^+/+^* cells again showed a significant increase of cytotoxicity against *Lgals9^-/-^* target cells compared to wild type YAC-1 cells, *Bst2^S/-^* NK cells showed slightly increased but statistically insignificant level of cytotoxicity increment against *Lgals9^-/-^* target cells compared to wild type YAC-1 target cells. *Bst2^-/-^* NK cells showed a modest increment of cytotoxicity against *Lgals9^-/-^* target cells compared to that against wild type YAC-1 target cells (Figure 5E). These results suggest that the short isoform of BST2 that lacks the first 12 amino acids (containing ITIM motif) of N-terminus in cytoplasmic tail does not participate in the inhibitory action of BST2 which is triggered by its interaction with Gal-9. In other words, the ITIM motif in the long isoform of BST2 is critical for the downregulation of NK cell cytotoxicity.

## 3. Discussion

BST2 is known as an anti-viral factor by physical interaction with budding virus and a modulator of type I interferon release by plasmacytoid dendritic cells [34]. However, in this study, we found that the cytoplasmic tail of BST2 had an ITIM motif, indicating that the cytotoxicity of NK cells could be regulated by BST2. To confirm this, we used various backgrounds of mice, such as B6 origins (*Bst2^+/+^*, *Bst2^+/-^*, or *Bst2^-/-^*), B6 x NZW F1 hybrids (*Bst2^S/+^* or *Bst2^S/-^*), and NZW backcrossed B6 mice (*Bst2^+/-^*, *Bst2^S/-^*, or *Bst2^-/-^*). We found that the cytotoxicity of *Bst2^-/-^* NK cells or short BST2 expressing (*Bst2^S/-^*) NK cells was greater than that of NK cells expressing the long isoform of BST2 (*Bst2^+/+^*, *Bst2^+/-^*, or *Bst2^S/+^*). We also found that Gal-9 and Gal-8 could be partners of BST2 to regulate NK activity. Blocking BST2 or Gal-9 with blocking monoclonal antibody efficiently increased cytotoxicity of long isoform BST2 expressing NK cells. However, it had little effect on NK cells lacking the long isoform of BST2. It is interesting since Gal-9 has been proposed as an immune regulatory molecule exerting its effect through TIM3 on cytotoxic cells [22]. In addition, Gal-8 blocking also showed similar results as Gal-9 blocking. This implies that several galectins could be potential signaling ligands of BST2. These results indicate that the presence of ITIM (SFYHYL) motif in BST2 is crucial for the regulation of NK cell cytotoxicity.

The N-terminal cytoplasmic tail of murine BST2 consists of 30 amino acids (20 amino acids in human BST2) and this N-terminal cytoplasmic tail is known to contain a hemITAM motif with a potential ITIM motif. Interestingly, BST2 can be alternatively translated at the second methionine (at 13th amino acid), resulting in a short isoform. Since the ITIM motif is located between the first methionine and the second methionine, the short isoform of BST2 does not contain the ITIM motif [3]. This fits well with our observations that only the long isoform shows inhibitory activity against NK cell cytotoxicity. NK cells might regulate the expression ratio of the long and the short isoforms of BST2 molecule depending on the activation status and inflammatory conditions, thereby regulating their cytotoxicity. Based on our observations that the presence of short isoform did not decrease the cytotoxicity of NK cells (Figure 5C,E), if BST2 could regulate NK cell cytotoxicity, the actual regulator would be the amount of the long isoform of BST2. We can speculate that signaling molecules delivering NK inhibitory signals downstream of BST2 interact with the long isoform cytoplasmic tail of BST2, but what exactly these signaling molecules are remains to be elucidated.

Galectins such as Gal-3 and Gal-9 are involved in the downregulation of NK cell cytotoxicity by binding to NKp30 and TIM3, respectively [22,23]. As NKp30 is an activating receptor, tumor-released Gal-3 can directly bind to NKp30 and block NKp30-mediated cytotoxicity of NK cells [23]. Gal-9 binds to TIM3, an immune check point molecule on NK cells, thus impeding NK cytotoxicity [22]. Interestingly, we observed a similar result regarding BST2/Gal-8 and 9-dependent downregulation of NK cell cytotoxicity. Blockade of either BST2 or Gal-8/Gal-9 by monoclonal antibodies enhanced the cytotoxicity of NK cells. Antibody-dependent cell-mediated cytotoxicity (ADCC) would not be the reason for the enhanced cytotoxicity after anti-BST2 and anti-Gal-8/-9 antibody treatment since the increased cytotoxicity was only observed in NK cells expressing the long isoform of BST2.

Considering the involvement of Gal-3 in the downregulation of NK cell activity, our findings suggest that galectins might be another line of NK cell inhibitory ligand systems that recruit diverse receptors such as NKp30, TIM3, and BST2, in addition to well-known receptor systems such as NKG2A and C in NK receptor complex (NKC) and killer cell immunoglobulin-like receptors (KIRs) in leukocyte receptor complex (LRC).

Many activating receptors on NK cells do not possess their own signaling motifs. Instead, they transmit the signal through noncovalently associated common adaptor molecules, such as DAP-10 or DAP-12, which contain ITAM motifs [35]. These adaptor molecules are not restricted to a specific activating receptor, therefore, controlling these adaptor molecules would be a clever way to regulate NK cell cytotoxicity initiated by various types of ligands. When NK cells attach to a target cell, an immunological synapse occurs at the interface between two cells [36]. Immune synapses are regions rich in lipid rafts where many NK cell receptors are located [37]. Both DAP-10 and DAP-12 are found at lipid rafts, in particular DAP-12 is reported to be found in its boundaries [38,39]. It is interesting that BST2 is also abundant in the boundaries of lipid rafts due to its topology [2,40]. The C-terminal GPI anchor of BST2 is embedded within the lipid raft region, whereas the N-terminus is located outside of lipid raft [2]. With this characteristic, BST2 can perform its function as an organizer of membrane microdomains [40].

DAP-12 has two crucial tyrosine residues in its ITAM motif. With strong signaling, both tyrosine residues are phosphorylated, and DAP-12 transmits the activation signal. However, when the signal is weak, only one tyrosine residue is phosphorylated and inhibitory signal is transmitted [27]. The partial sequence of DAP-12 ITAM motif has SDVYSDL(SxxYxxL), which coincidently resembles with the sequence in ITIM of BST2; SYDYCRV(SxxYxxV). It might be possible that BST2 somehow interfere NK cell adaptor molecules such as DAP-12 by direct or indirect interactions in the lipid raft. Another possibility would be the competition between BST2 and DAP12 for the phosphorylation of signaling motifs, thus resulting in the regulation of NK cell cytotoxicity.

NK cells were first chosen to be analyzed in this study. However, the expression of BST2 in T cells or B cells was also increased after IL-2 stimulation. Thus, the regulatory mechanism through BST2 could be operational in these cell types (Appendix A). To test whether BST2 functioned similarly in cytotoxic CD8+ T cells, we introduced *Bst2^-/-^* into *OT1* mice harboring ovalbumin- specific T cell receptor transgene. Antigen stimulated *OT1* showed increased expression of BST2. *Bst2^-/-^ OT1* cells showed higher cytotoxicity against ovalbumin expressing E.G7 target cells than *Bst2^+/+^ OT1* cells. Neither *Bst2^+/+^* nor *Bst2^-/-^ OT1* cells showed any cytotoxicity against mother cell line EL4, which did not express ovalbumin (Appendix A). This result suggests that BST2 possibly can act as an inhibitory receptor in cytotoxic T cells in addition to NK cells.

In this study, we found that the long isoform of BST2 could transmit inhibitory signals through the cytoplasmic tail containing an ITIM motif by recognizing Gal-8 and Gal-9. This suggests that galectins are new self-inhibitory ligands to modulate the over-stimulation of NK cell activity. Based on the results of previous reports suggesting the association of galectins with tumor malignancy in breast cancer, renal cancer, thyroid cancer, and other cancers [41], antibody therapy aiming at blocking either BST2 or Gal-8/-9 could be an option for anti-cancer therapy.

## 4. Materials and Methods

### 4.1. Cell Lines and Mice

Tumor cell lines YAC-1 (ATCC TIB-160), RMA-S, B16, EL4 (ATCC TIB-39), and E.G7-OVA (ATCC CRL-2113) were cultured in Roswell Park Memorial Institute (RPMI) 1640 Medium (RPMI 1640, Gibco) supplemented with 10% fetal bovine serum (FBS), 2 mM L-glutamine, 1% penicillin-streptomycin, 10 μg/mL gentamycin, and 50 µM β-mercaptoethanol (Gibco-BRL). *Bst2* knockout mice (C57BL/6 background) were created on a C57BL/6Tac background by Xenogen Biosciences (Cranbury, NJ). NZW/N mice expressing cytoplasmic tail deficient, short isoform of BST2 (*Bst2^S/S^*) were purchased from Japan SLC (Hamamatsu, Japan) and crossed with *Bst2* knockout C57BL/6 (B6) mice to generate B6/NZW F1 mice. To generate C57BL/6 mice containing *Bst2* gene of NZW origin, NZW/N mice were backcrossed with C57BL/6 *Bst2^-/-^* mice at least eight times. All animal experiment protocols of this study were approved by the Institutional Animal Care and Use Committee of Korea University (KUIACUC-2018-25, 29 March~31 December 2018; and KUIACUC-2019-0003, 1 January 2019~31 December 2020).

### 4.2. Cell Line Gene Knock out

Murine Gal-9 genes (*Lgals9*) of YAC-1 were knocked out with a CRISPR-Cas9 system. Target sequence of *Lgals9* used was 5′- CACGAAGCAGAACGGACAGT-3′. These target sequences were cloned into pX330 vector to make gene-specific gRNA. Reporter vector (tagged with GFP) was co-transfected. At two days after transfection, GFP positive cells were sorted and spread as single cells in a 96-well plate. Among single cells, clones with frame-shift mutation on its both alleles were selected as gene knocked out cells.

### 4.3. Preparation of NK Cells

For the in vivo stimulation and preparation of NK cells, 150 µg of Poly I:C was intraperitoneally injected. At 16 h after injection, mice were sacrificed and splenic NK cells were isolated using DX-5 microbeads (Miltenyi Biotec, Bergisch Gladbach, Germany).

For in vitro stimulation of NK cells, Naïve splenic NK cells were isolated using DX-5 microbeads. 1 × 10^6^/mL of sorted cells were cultured with 50 µg/mL of Poly I:C and 200 U/mL of IL-2 (PeproTech, Cranbury, NJ, USA) for 16 h. 

For the generation of IL-2 activated NK cells (also known as Lymphokine Activated Killer cells or LAK cells), 4 × 10^6^ cells/mL of whole splenocytes were cultured in a 100 mm dish with 1000 U/mL of IL-2 for 72 h. Plate unbounded cells were removed and the remaining cells on the plate were further incubated with a 1:1 mixture of fresh medium containing 500 U/mL of IL-2 and conditioned medium for four days. These cells were composed of over 80% NK cells (NK1.1^+^TCRβ^−^ cells).

### 4.4. Preparation of OT1 Cells

CD8α^+^ splenocytes from *OT1* mice were sorted using anti-CD8α microbeads (Miltenyi Biotech, Ly-2) and Magnetic-activated cell sorting (MACS) column (Miltenyi Biotech). Sorted cells presented TCRvα2 and TCRvβ5 exclusively that could specifically bind to K^b^-OVA_257–264_. Naïve *OT1* cells were stimulated with biotin-conjugated K^b^-OVA/anti-CD28 antibody (eBioscience, San Diego, CA, USA, 37.51), which were mixed and bound to anti-biotin MACSiBead particles (Miltenyi Biotech). They were then incubated with 30 units/mL of IL-2 in RPMI medium for 2 to 3 days. Magnetic beads were removed with a magnetic bar prior to any assay.

### 4.5. Cytotoxic Assay

Before incubating with antibodies, FcγRs were blocked with anti-CD16/32 antibody (2.4G2) for 15 min at 4 °C. Then 10 μg/mL of functional grade anti-BST2 antibody (eBioscience, eBio927) and 50 μg/mL papain-treated anti-galectin antibody (IsuAbxis, Gyeonggi-do, Korea) were used to block respective antigens.

For 7-AAD/CFSE assay, NK cells were labeled with 0.625 μM of CFSE at 37 °C for five minutes. Next, NK cells and target cells were co-cultured for four hours. After 0.5 μg/mL of 7-AAD (BioLegend, San Diego, CA, USA) was added, the mixture was incubated at room temperature for 15 min. Cells were then fixed with 1% paraformaldehyde and analyzed by flow cytometry [42]. Finally, cytotoxicity was calculated using the following formula [43]:% cytotoxicity = [(test release) − (spontaneous release)]/[(maximal release) − (spontaneous release)] × 100 

For Calcein-AM release assay, tumor cells were labeled with 15 μg/mL of Calcein-AM (Thermo Fisher Scientific) for two hours in a 37 °C incubator. Non-fluorescent calcein-AM would turn green after removal of its acetoxymethyl group by intracellular esterases of live cells. Effector cells and tumor cells were co-cultured for one to four hours. After centrifuging the plate at 300× *g* for 4 min at room temperature, supernatants were collected to detect released as green-fluorescent calcein from tumor cells. Fluorescence was measured at the maximum excitation and emission wavelengths at 485 nm and 535 nm, respectively, using a SpectraMAX I3x. The maximum released calcein was determined after lysing target cells with 3% of Triton X-100. To calibrate the effect of Triton X-100 on results, medium containing triton was added (mediaTriton). The cytotoxicity was then calculated using the following formula:% Specific lysis = % cytotoxicity = [(test release) − (spontaneous release)]/[{(maximal release) − (media + Triton)} − {(spontaneous release) − (media only)}] × 100

### 4.6. Flow Cytometry

For flow cytometry staining, 0.5–1 × 10^6^ cells were used. Non-specific binding of antibody to FcγR was blocked with anti-CD16/32 antibody (2.4G2) for 15 min on ice. Cells were incubated with fluorochrome-conjugated antibodies against cell surface antigen and then washed with flow cytometry buffer [PBS with 0.1%BSA (Sigma-Aldrich) and 0.01% sodium azide]. Fixation, permeabilization and staining of intracellular antigen was performed after surface staining. The following monoclonal antibodies were used: fluorescein isothiocyanate (FITC) conjugated anti-B220 (BD Bioscience, RA3-6B2), CD11b (BD Bioscience, M1/70), TCRβ (BD Bioscience, H57), CD25 (BD Bioscience, 7D4), NKG2A/C/E (BD Bioscience, 20d5), CD107a (BD Bioscience, 1D4B), CD44 (BD Bioscience, IM7), phycoerythrin (PE) conjugated streptavidin (BD Bioscience, 2B4), anti-NK1.1 (BD Bioscience, PK136), NKp46 (BD Bioscience, 29A1.4), CD27 (BioLegend, LG.3A10), Granzyme B (eBioscience, NGZB), CD8α (BD Bioscience, 53–6.7), PerCP-Cy5.5 conjugated anti-CD11c (BD, HL3), NK1.1 (BD Bioscience, PK136), eFluor660 conjugated anti-T-bet (eBioscience, ebio4B10), PerCP-eFluor710 conjugated anti-Ly49G2 (eBioscience, ebio4D11), allophycocyanin (APC) conjugated anti-NK1.1 (BD Bioscience, PK136), TCRβ (BD Bioscience, H57), NKG2D (BD Bioscience, CX5), CD62L (BD Bioscience, MEL-14), PE-Vio770 conjugated anti-CD3 (Miltenyi, REA641), APC-Cy7 conjugated anti-CD11b (BD, M1/70), BV421 conjugated anti- IFN-γ (BD, XMG1.2), BV786 conjugated anti-TCR β (BD, H67-597), PE-Cy7 conjugated anti-KLRG1 (eBioscience, 2F1) and biotin conjugated anti-PDCA-1 (eBioscience, e.Bio927). For CD107a staining, anti-CD107a antibody and Golgistop (BD, containing monensin) were added directly to cell culture media where effector cells and target cells were co-cultured. Flow cytometric data were acquired on an FACS Calibur, Verse, Lyric and Fortessa (Becton Dickinson). Data were analyzed using FlowJo version 10 (FlowJo, LLC)

### 4.7. Enzyme-Linked Immunosorbent Assay (ELISA)

After cytotoxicity assays were completed, culture supernatants were harvested for the detection of Granzyme B. Quantification was completed using ELISA kit (R&D systems, Minneapolis, MN, USA, DY1865) following the manufacturer’s protocol. Briefly, we coated ELISA plate with capture antibody for granzyme B and blocked the plate with BSA solution. After blocking, culture supernatants were added and incubated for appropriate time. Detection antibody for granzyme B was treated in order. Finally, we digitalized the amount of granzyme B in the plate after adding TMB solution followed by absorbance measurement with a microplate reader (Bio-Rad model 680).

### 4.8. Pull-Down Assay and Western Blot Analysis

Recombinant BST2 was obtained from plasmid transfected 293T cells as described previously [44]. GST-tagged fusion proteins Gal-8, Gal-9, and Gal-3 were expressed in *Escherichia coli* (*E. coli)* and then purified by glutathione affinity resin [45]. GST-tagged galectins were immobilized on glutathione-agarose beads and incubated with recombinant BST2. Protein complexes were then eluted with elution buffer from beads. Next, proteins were resolved by sodium dodecyl sulfate-polyacrylamide gel electrophoresis (SDS-PAGE) and silver stained. To confirm the interaction with BST2 and galectins, proteins were transferred to nitrocellulose membranes. Blots were incubated with rabbit anti-BST2 polyclonal antibody and detected with horseradish peroxidase (HRP)-conjugated anti-rabbit IgG. To detect galectin-9 in each target tumor cell lines, cells were lysed with M-PER (Mammalian Protein Extraction Reagent, Thermo scientific) with proteinase inhibitor. The concentration of total proteins was measured by Bradford assay (Bio-Rad, 5000006). 15μg of proteins per lane were loaded and separated by SDS-PAGE and transferred to PVDF (Polyvinylidene fluoride) membrane (Bio-Rad, 1620177). Purified anti-mouse galectin-9 antibody (Biolegend, 108A2, 137901) was used for primary antibody and HRP-conjugated goat anti-rat IgG (Santa Cruz Biotechnology, sc-2006) was used for detection antibody. Anti-GAPDH antibody (Santa Cruz Biotechnology, 6C5, sc-32233) and HRP-conjugated anti-mouse IgG (Sigma-Aldrich) were used for detection of GAPDH, a housekeeping gene. Blot images were obtained using an LAS4000 mini (GE Healthcare, Chicago, IL, USA).

### 4.9. Statistical Analysis

GraphPad Prism Version 6.0 (GraphPad Software Inc., San Diego, CA, USA) was used to perform all statistical analyses. Differences in data were compared using unpaired two-tailed t-test or two-way analysis of variance (ANOVA) with Sidak’s multiple comparison test. Data are expressed as mean ± standard deviation (S.D.). Differences were considered significant if *p* < 0.05. Asterisk was used to indicate significance levels (*, *p* < 0.05; **, *p* < 0.01; ***, *p* < 0.001; ****, *p* < 0.0001; n.s., not significant).

## Figures and Tables

**Figure 1 ijms-23-11395-f001:**
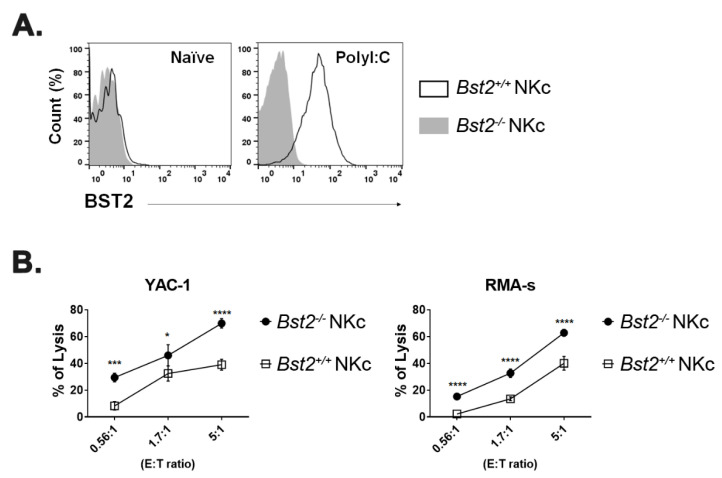
PolyI:C stimulated *Bst2^-/-^* NK cells have a higher cytotoxicity to tumor cells than *Bst2^+/+^* NK cells. B6-origin *Bst2* wildtype (+/+) and *Bst2* knockout (-/-) mice were intraperitoneally injected with polyI:C. At 16 hrs after injections, splenic NK cells were isolated using DX-5 microbeads. (**A**) *Bst2^+/+^* and *Bst2^-/-^* NK cells were stained with anti-BST2 antibody and analyzed by flow cytometry. NK1.1^+^ TCRβ^-^ cells were indicated as NK cells. Data are representative of two independent experiments. (**B**) Calcein-AM stained target cells (YAC-1 and RMA-S) were co-cultured with *Bst2^+/+^* and *Bst2^-/-^* NK cells for 4 hrs at 37 °C in a humidified incubator. Released calcein was measured by SpectraMAX with excitation/emission wavelengths at 485 nm/535 nm. Graphs showed a mean ± S.D. (*n* = 3 per group). Data are representative of three independent experiments. (*, *p* < 0.05; ***, *p* < 0.001; ****, *p* < 0.0001).

**Figure 2 ijms-23-11395-f002:**
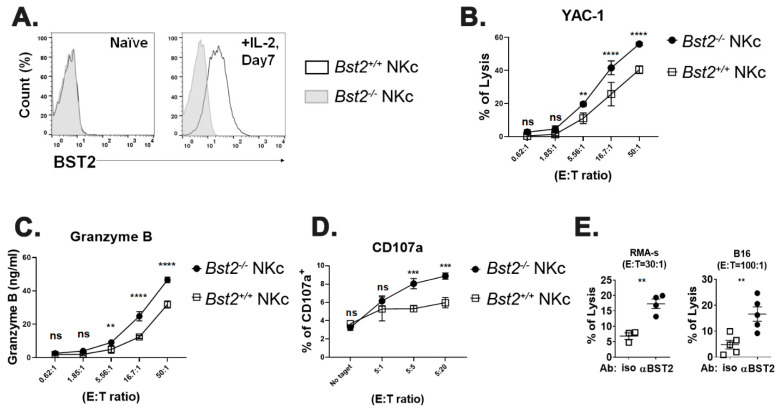
*Bst2^-/-^* lymphokine activated killer cells have higher cytotoxicity to tumor cells than *Bst2^+/+^* LAK cells. Naïve splenocytes of B6-origin *Bst2^+/+^* and *Bst2^-/-^* mice were stimulated with IL-2 for seven days to generate LAK cells. (**A**) *Bst2^+/+^* and *Bst2^-/-^* LAK cells were stained with anti-BST2 antibody and analyzed by flow cytometry. (**B**) Calcein-AM stained target cells (YAC-1) were co-cultured with *Bst2^+/+^* and *Bst2^-/-^* LAK cells for 1 hr at 37 °C in a humidified incubator. Released calcein was measured to analyze cytotoxicity of effector cells as described in Figure 1. Graphs showed a mean ± S.D. (*n* = 3–4 per group). Data are representative of three independent experiments. (**C**) After detecting calcein released in media, released granzyme B in the same media was measured by enzyme-linked immunosorbent assay (ELISA). Graphs showed a mean ± S.D. (*n* = 3–4 per group). Data are representative of two independent experiments. (**D**) *Bst2^+/+^* and *Bst2^-/-^* LAK cells were co-cultured with target cells (YAC-1) in culture media containing Golgistop and anti-CD107a antibody. NK1.1 and TCRβ were stained after 1 hr of cultivation at 37 °C in a humidified incubator. Percentages of CD107a positive cells from LAK cells (NK1.1^+^ TCRβ^-^) were indicated. Graphs showed a mean ± S.D. (*n* = 3 per group). Data are representative of two independent experiments. (**E**) CFSE stained *Bst2^+/+^* LAK cells were treated with isotype control or anti-BST2 antibodies and co-cultured with target cells (RMA-S and B16) for 4 hrs at 37 °C in a humidified incubator. Cells were stained with 7-AAD after cultivation. These cells were then fixed with 1% paraformaldehyde and analyzed by flow cytometry. CFSE^-^, 7-AAD^+^ population is designated as lysed target cells. The word ‘iso’ indicates isotype control antibody treatment and ‘αBST2′ indicates anti-BST2 antibody treatment. Graphs showed a mean ± S.D. (*n* = 3–5 per group). (**, *p* < 0.01; ***, *p* < 0.001; ****, *p* < 0.0001; n.s., not significant).

**Figure 3 ijms-23-11395-f003:**
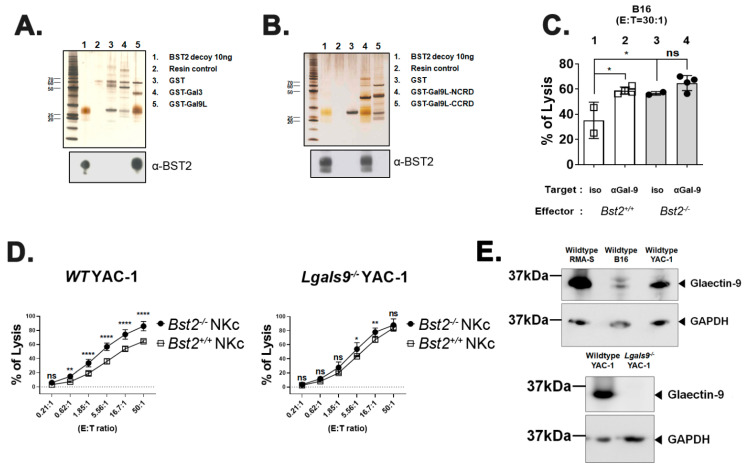
Galectin-9 is associated with BST2-dependent NK cell activity. (**A**) Recombinant BST2 protein was pulled down with GST-tagged Gal-3 (GST-Gal3), as well as the GST-tagged long isoform of Gal-9 (GST-Gal9L). Precipitates were then analyzed by western blotting with anti-BST-2 antibody. (**B**) Recombinant BST2 protein was pulled down with either GST-tagged NCRD or CCRD of Gal9L (GST-Gal9L-NCRD or GST-Gal9L-CCRD). Precipitates were then analyzed by western blotting with anti-BST-2 antibody. (**C**) Naïve splenocytes of B6-origin *Bst2^+/+^* and *Bst2^-/-^* mice were stimulated by IL-2 for seven days to generate LAK cells. CFSE stained *Bst2^+/+^* and *Bst2^-/-^* LAK cells were co-cultured with target cells (B16) for 4 hrs at 37 °C in a humidified incubator in the presence of isotype control antibody or anti-Gal-9 antibody. Cells were stained with 7-AAD after cultivation. These cells were then fixed with 1% paraformaldehyde and analyzed by flow cytometry. CFSE^-^, 7-AAD^+^ population is designated as lysed target cells. Graphs showed a mean ± S.D. (*n* = 2–4 per group). (**D**) Naïve splenocytes of B6-origin *Bst2^+/+^* and *Bst2^-/-^* mice were stimulated by IL-2 for seven days to generate LAK cells. Calcein-AM stained target cells (Wild type (WT) YAC-1 and *Lgals9^-/-^* YAC-1) were co-cultured with *Bst2^+/+^* and *Bst2^-/-^* LAK cells for 3 hrs in a 96-well round plate. Released calcein was measured to analyze cytotoxicity of effector cells as described in Figure 1. Graphs showed a mean ± S.D. (*n* = 4–6 per group). Data are representative of two independent experiments. (**E**) Western blotting was performed on whole cell lysates of tumor target cell lines with anti-galectin-9 antibody and anti-GAPDH antibody. GAPDH was used as a loading control. The upper panel showed the result of wildtype tumor cell lines (RMA-S, B16, and YAC-1) and the lower panel showed the comparison of wildtype YAC-1 and *Lgals9^-/-^* YAC-1 clone used in this study. (*, *p* < 0.05; **, *p* < 0.01; ****, *p* < 0.0001; n.s., not significant).

**Figure 4 ijms-23-11395-f004:**
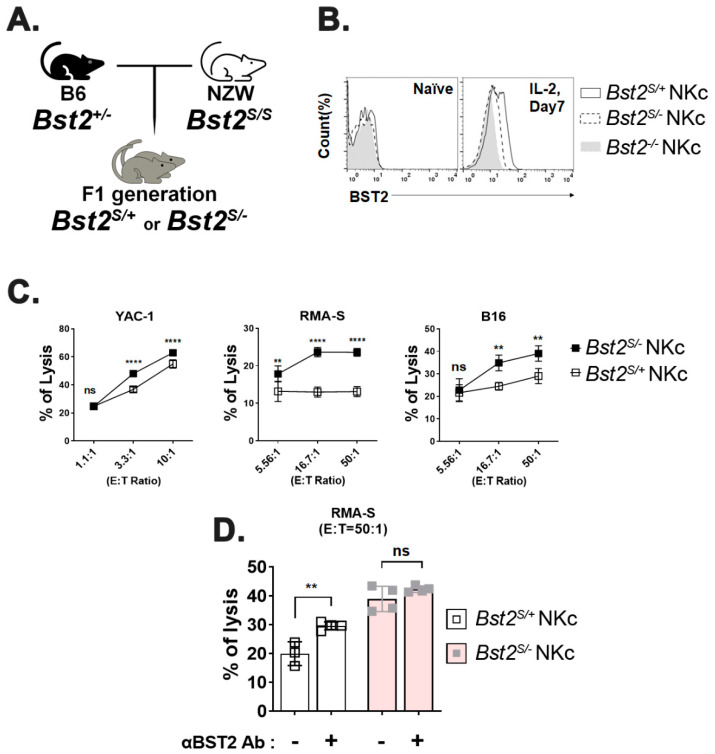
Long isoform, but not short isoform, of BST2 on NK cells impedes cytotoxicity. (**A**) F1 hybrids were generated by crossing B6-origin *Bst2^+/-^* mice and NZW mice (*Bst2^S/S^*). (**B**) Naïve splenocytes of F1 hybrids and B6-origin *Bst2^-/-^* mice were stimulated by IL-2 for seven days to generate LAK cells. Expression of BST2 was detected through flow cytometry staining. (**C**) CFSE stained *Bst2^S/+^* and *Bst2^S/-^* LAK cells were co-cultured with target cells (YAC-1, RMA-S, and B16) for 4 hrs at 37 °C in a humidified incubator. Cells were stained with 7-AAD after cultivation. These cells were then fixed with 1% paraformaldehyde and analyzed by FACS. CFSE-, 7-AAD+ population is designated as lysed target cells. Graphs showed a mean ± S.D. (*n* = 4 per group). (**D**) CFSE stained *Bst2^S/+^* and *Bst2^S/-^* LAK cells from F1 hybrids were co-cultured with target cells (RMA-S) cells for 4 hrs at 37 °C in a humidified incubator in absence or presence of anti-BST2 antibody. Graphs showed mean ± S.D. (*n* = 3–4 per group). (**, *p* < 0.01; ****, *p* < 0.0001; n.s., not significant).

**Figure 5 ijms-23-11395-f005:**
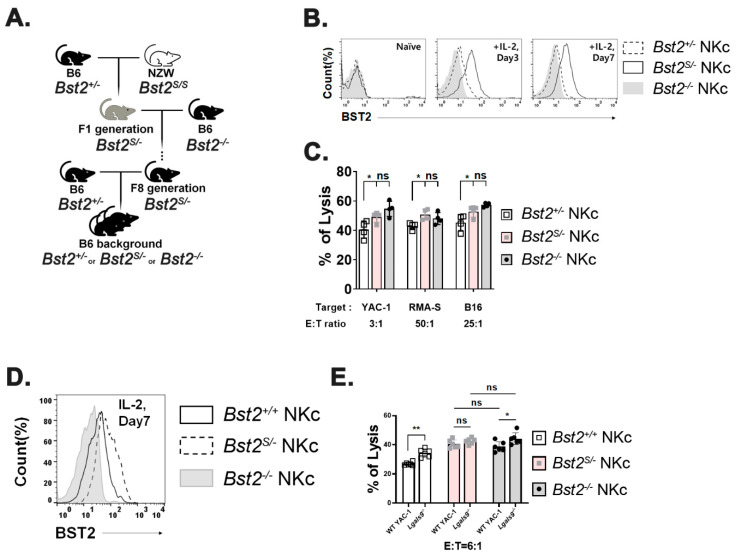
NK cells with short isoform of BST2 show enhanced cytotoxicity as well as BST2 deficient NK cells. (**A**) Backcross from NZW (*Bst2^S/S^*) background to B6 background. (**B**) BST2 expression of LAK cells generated from splenocytes of B6-background *Bst2^+/-^*, *Bst2^S/-^*, and *Bst2^-/-^* mice. (**C**) Calcein-AM stained target cells (YAC-1, RMA-S, and B16) were co-cultured with *Bst2^+/-^*, *Bst2^S/-^*, and *Bst2^-/-^* LAK cells for 4 hrs at 37 °C in a humidified incubator. Released calcein was measured to analyze cytotoxicity of effector cells as described in Figure 1. Data are representative of two independent experiments. Graphs showed mean ± S.D. (*n* = 4 per group). (**D**) Naïve splenocytes of B6-origin *Bst2^+/+^*, *Bst2^S/-^*, and *Bst2^-/-^* mice were stimulated by IL-2 for seven days to generate LAK cells. Expression level of BST2 was measured through flow cytometry staining. (**E**) Calcein-AM stained target cells (YAC-1 and *Lgals9^-/-^* YAC-1) were co-cultured with *Bst2^+/+^*, *Bst2^S/-^*, and *Bst2^-/-^* LAK cells for 4 hrs at 37 °C in a humidified incubator. Released calcein was measured to analyze cytotoxicity of effector cells as described in Figure 1. Graphs showed a mean ± S.D. (*n* = 6 per group). (*, *p* < 0.05; **, *p* < 0.01; n.s., not significant).

**Table 1 ijms-23-11395-t001:** ITIM motif of the cytoplasmic tail domain in BST2.

Gene	Species	Amino acid Sequence Alongside ITIM Motif	ITIM Tyrosine Residue	Reference
*BST2*	Human	MASTSYDYCRVPMEDGDKRC…	Y8	
*Bst2*	Mouse	MAPSFYHYLPVPMDEMGGKQGWGS…	Y6	
*Hs1bp3*	Mouse	…GHVEYQILVVTR…VSKKYSEIEEFYQKLSSRY…	Y41, Y71, Y78	[28]
*CD33*	Human	…ELHYASLNFH…STEYSEVRTQ	Y340, Y358	[29]

1. Underline indicates ITIM motif. 2. Red character indicates conserved amino acids of ITIM motif.

## Data Availability

The data presented in this study are available in article or Appendix A.

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
