# Peer review of "BST2, a Novel Inhibitory Receptor, Is Involved in NK Cell Cytotoxicity through Its Cytoplasmic Tail Domain"

_ijms, 2022, doi:10.3390/ijms231911395_

Round 1

Reviewer 1 Report

In this study, Oh and colleagues describe the regulatory role of BST2 inhibitory receptor in NK cell cytotoxic function against tumor cells. To exert its inhibitory function, the involvement of cytoplasmic tail domain of BST2 were also shown. Galectin-9 and -8 are proposed as potential ligands for BST2 to inhibit NK cell activation. The key findings in this study are novel and interesting, however, many of the important information are lacking in the presented manuscript to support their conclusion drawn.

1. Three target tumor cell lines were used in this study, YAC-1, RMA-S and B16. Considering they are vastly different in their expression of both activating ligands and inhibitory ligands of NK cells, it is absolutely crucial to characterize their expression of Gal-9 and Gal-8.

2. It is quite odd there is no mention of how BST2 inhibitory receptor can be involved in regulating other effector functions of NK cells than cytotoxicity. Specifically, it is also crucial to demonstrate the role of BST2 in NK cell IFN-gamma production. In this context, we need to know whether Gal-9 and -8 are also involved in regulating IFN-gamma production of NK cells.

3. In line with my comment #1, the KO efficacy of Gal-9-deficient YAC-1 cells must be shown. Additionally, why YAC-1 cell line was used for making KO cells?

4. Considering the authors’ claim that BST2 inhibits NK cell function through its cytoplasmic ITIM domain, it must inhibit NK cell activating signal without any preferences. NK cells can be activated through NKG2D (YAC-1), DNAM-1 (B16), or unspecified (RMA-S) receptor recognition, or even cytokine stimulation (IL-12). Role of BST2 inhibitory receptor against those different activating signal in NK cells should be addressed.

4. The data presented as Supp Fig 3D must be important and only one data to support Gal-8 as a potential BST2 ligand, therefore why not presented as a main Figure data?

5. Why does all cytotoxicity data so vary between the experiments shown in this study? For instance, in Figure 4, C panel S/+ RMA-S killing data shows 12-13% lysis in all ET ratio, but in D panel goes to 20% (although we do not know ET ratio of this data). There are many of those larger inconsistencies of killing data in this work, so please explain the reason.

6. As mentioned above in my comment #5, there are no ET ratio information in the Figures, Fig 3C, Supp Fig 3D, Fig 4D. Those need to be indicated.

Author Response

Thank you for your valuable comments. The answers and figures to the question have been uploaded as attached PDF file. For convenience, we also write the full text here.

Point 1: Three target tumor cell lines were used in this study, YAC-1, RMA-S and B16. Considering they are vastly different in their expression of both activating ligands and inhibitory ligands of NK cells, it is absolutely crucial to characterize their expression of Gal-9 and Gal-8.

Response 1: Yes, galectin-9 expression varies by cell type. Basically, galectins are kind of secretory molecule. So we detected galectin-9 expression by western blotting method. We found that all target tumor cell lines used in this study express galectin-9, although their expression levels were quite different. Of note, RMA-S also expressed galectin-9 on its surface.

Point 2: It is quite odd there is no mention of how BST2 inhibitory receptor can be involved in regulating other effector functions of NK cells than cytotoxicity. Specifically, it is also crucial to demonstrate the role of BST2 in NK cell IFN-gamma production. In this context, we need to know whether Gal-9 and -8 are also involved in regulating IFN-gamma production of NK cells.

Response 2: We plan to elucidate the detailed mechanism of BST2 on NK cell inhibition. We cannot provide a definite answer to this question at this time, but further research directions will be discussed in Response 4. We also analyzed intracellular levels of IFN-gamma in WT and KO NK cells with and without stimulation. We found that intracellular levels of IFN-gamma increased after polyI:C stimulation, but still there was no difference between WT and KO NK cells. Levels of IFN-gamma increased after meeting the target cells (in this case, YAC-1 cells). Although there were some fluctuations, still no differences were found between WT and KO NK cells. Nevertheless, we cannot be sure that BST2 does not affect the production of IFN-gamma.

Point 3: In line with my comment #1, the KO efficacy of Gal-9-deficient YAC-1 cells must be shown. Additionally, why YAC-1 cell line was used for making KO cells?

Response 3: We made two independent monoclones of the galectin-9 knock-out YAC-1. These two clones showed complete depletion of galetin-9 expression and were used as target cells in this study. Since YAC-1 cells were used as a main target cells for NK cell cytotoxicity test throughout this study, we decided to knocked out galectin-9 gene in this cell line to maintain consistency of data analysis.

Point 4: Considering the authors’ claim that BST2 inhibits NK cell function through its cytoplasmic ITIM domain, it must inhibit NK cell activating signal without any preferences. NK cells can be activated through NKG2D (YAC-1), DNAM-1 (B16), or unspecified (RMA-S) receptor recognition, or even cytokine stimulation (IL-12). Role of BST2 inhibitory receptor against those different activating signal in NK cells should be addressed.

Response 4: Thanks for the critical comment. We are currently actively searching for answer(s) as an independent study. One of the ideas we are currently standing on is that most activating receptors on NK cells do not possess their own signaling motifs. Instead, they transmit the signal through common adapter molecules such as DAP-10 or DAP-12 which contain ITAM motifs. These adapter molecules are not restricted to a specific activating receptor. Therefore, controlling these adapter molecules would be a clever way to regulate NK cell cytotoxicity initiated by various types of ligands. When NK cells attach to a target cell, an immunological synapse occurs at the interface between the two cells. Immune synapses are regions rich in lipid raft, and lipid rafts are where many NK cell receptors are located. DAP-10 and DAP-12 both are found at lipid raft boundaries. It is really interesting that BST2 is also abundant in lipid raft boundaries due to its topology. The N-terminus of BST2 is in non-lipid raft region, whereas the C-terminal GPI anchor of BST2 is embedded in lipid raft. By its function, BST2 is known as an organizer of membrane microdomains. Another idea is the possibility of competition between BST2 and adapter molecules in the recruitment of signaling molecules. DAP-12 has two crucial tyrosine residue in its ITAM motif. With strong signaling, both tyrosine residues are phosphorylated and DAP-12 transmits the activation signal. However, when the signal is weak and only one tyrosine residue is phosphorylated, an inhibitory signal is transmitted. The partial sequence of DAP-12 ITAM motif has SDVYSDL(SxxYxxL), which coincidently resembles with the sequence in ITIM of BST2; SYDYCRV(SxxYxxV). Although we do not have critical experimental data at the moment, many circumstantial evidence inspires that BST2 may affect the action of adaptor molecules. Based on these previous findings, we are currently investigating exactly which molecule(s) interacts with BST2 at the cellular level.

Point 4: The data presented as Supp Fig 3D must be important and only one data to support Gal-8 as a potential BST2 ligand, therefore why not presented as a main Figure data?

Response 4: Unfortunately, the data set for galectin-8 is not sufficient compared to the set of galectin-9, so we mainly focused on galectin-9 in the text. Although we considered galectin-8 was also a potential ligand of BST2, further investigation into galectin-8 are needed to provide other evidence for galectin-8 involvement. For now, this is not yet complete and we have left galectin-8 as supplemental data.

Point 5: Why does all cytotoxicity data so vary between the experiments shown in this study? For instance, in Figure 4, C panel S/+ RMA-S killing data shows 12-13% lysis in all ET ratio, but in D panel goes to 20% (although we do not know ET ratio of this data). There are many of those larger inconsistencies of killing data in this work, so please explain the reason.

Response 5: Cytotoxicity is the result of the interaction of effector cells with target cells, meaning that the state of target cells can also affect cytotoxicity. Of course, it would be ideal to keep consistent condition, but there were some deviations due to practical errors. However, these variations do not distort the function of BST2 as the condition of target cells within individual experiment were the same for both WT and KO effector cells.

Point 6: As mentioned above in my comment #5, there are no ET ratio information in the Figures, Fig 3C, Supp Fig 3D, Fig 4D. Those need to be indicated.

Response 6: Missing indicators are inserted for Fig 3C, Supp Fig 3D and Fig 4D.

Reviewer 2 Report

Although the findings that BST2 were involved in natural killer cell cytotoxicity through its cytoplasmic tail domain are very interesting, numbers of points need clarifying and certain statements require further justification. These are given below.

<Points>

1.      “BST2” is also called “tetherin” or “CD317”. For the benefit of readers, please add another name(s) in introduction. 

2.      The authors described, “All animal experiment protocols of this study were approved by the Institutional Animal Care and Use Committee of Korea University (KUIACUC-2018-259)” without showing approval date (lines 362-364). Please show the approval date.

3.      In figure legends, how many independent experiments were performed should be added. For example, “n=?”.

4.      Reference style is not “IJMS” style. 

5.      In line 113, “WT” and “KO” should be changed to “wild type” and “knockout”, respectively.

6.      In line 358, “Xenogen Biosciences (USA)” should be changed to “Xenogen Biosciences (Cranbury, NJ)”.

7.      In line 259, “Japan SLC” should be changed to “Japan SLC (Hamamatsu, Japan)”.

Author Response

Thank you for your valuable comments. The answer to the question have been uploaded as attached PDF file. For convenience, we also write the full text here.

Point 1: “BST2” is also called “tetherin” or “CD317”. For the benefit of readers, please add another name(s) in introduction.

Response 1: We changed first sentence in introduction from “BST2 restricts the release of viruses at the plasma membrane” to “Bone Marrow Stromal Cell Antigen 2 (BST2; also known as tetherin, CD317, PDCA-1 and HM1.24) restricts the release of viruses at the plasma membrane”

Point 2: The authors described, “All animal experiment protocols of this study were approved by the Institutional Animal Care and Use Committee of Korea University (KUIACUC-2018-259)” without showing approval date (lines 362-364). Please show the approval date.

Response 2: We updated and changed sentence from “(KUIACUC-2018-25)” to “(KUIACUC-2018-25, 2018-03-29 ~ 2018-12-31; and KUIACUC-2019-0003, 2019-01-01 ~ 2020-12-31).”

Point 3: In figure legends, how many independent experiments were performed should be added. For example, “n=?”.

Response 3: We clarified figure legends and add information about technical replicates.

Point 4: Reference style is not “IJMS” style.

Response 4: We updated reference style properly. Sorry for inconvenience, it is now IJMS style.

Point 5: In line 113, “WT” and “KO” should be changed to “wild type” and “knockout”, respectively.

Response 5: We changed Line 113 from “B6-origin Bst2 WT (+/+) and Bst2 KO (-/-) mice were intraperitoneally injected with polyI:C.” to “B6-origin Bst2 wildtype (+/+) and Bst2 knockout (-/-) mice were intraperitoneally injected with polyI:C.” And other sentences containing ‘WT’ or ‘KO’ were also changed to ‘wild type’ or ‘knockout’, respectively.

Point 6: In line 358, “Xenogen Biosciences (USA)” should be changed to “Xenogen Biosciences (Cranbury, NJ)”.

Response 6: We changed Line 358 from “Bst2 knock out mice (C57BL/6 background) were created on a C57BL/6Tac background by Xenogen Biosciences (USA)” to “Bst2 knock out mice (C57BL/6 background) were created on a C57BL/6Tac background by Xenogen Biosciences (Cranbury, NJ).”

Point 7: In line 259, “Japan SLC” should be changed to “Japan SLC (Hamamatsu, Japan)”.

Response 7: We changed Line 359 from “short isoform of BST2 (Bst2S/S) were purchased from Japan SLC” to “short isoform of BST2 (Bst2S/S) were purchased from Japan SLC (Hamamatsu, Japan).”

Reviewer 3 Report

The article by Oh et al. “BST2, a novel inhibitory receptor, is involved in NK cell cytotoxicity through its cytoplasmatic tail domain” describes BST2 in the control of NK cell cytotoxicity in murine models. The authors use different transgenic mouse models to evaluate the function of BST2 and its ITIM motif in the cytoplasmic tail. NK cells derived from BST2 knock out mice show enhanced cytotoxicity against YAC-1 and RAM-S cell lines compared to NK cells expressing BST2. In line with these results, BST2 expressing NK cells incubated with a blocking antibody against BST2 enhanced the NK cytotoxic activity as well. Furthermore, using NZW mice which express ITIM-deficient BST2 showed higher cytotoxicity than wildtype NK cells. Finally, the authors performed pulldown assays to determine galectin-8 and -9 as potential ligands for BST2. 

So far, no studies have described BST2 as an inhibitory NK receptor recognizing galectin-8 and -9. Galectins are known to be associated with tumor growth and metastasis and therefore identifying blocking antibodies either targeting BST2 or galectins might represent an attractive target for cancer therapy to enhance NK cell cytotoxicity. 

1.)   Figure 2C: Please indicate the amount of granzyme B in pg/ml or ng/ml instead of absorbance. 

2.)   Describe according to table 1 why Hs1bp3 or Cd33 were choicen for comparison. Both proteins are not mentioned in the text at all. 

3.)   Explain in the result section why polyI:C was used as treatment for BST2 expression. 

4.)   Please introduce the abbreviation BST2 in the abstract and main text. 

5.)   Line 54: Please change “…can recognize a variety of antigens.” To “a variety of ligands” and give some examples of NK ligands. 

6.)   Line 55: add death receptor-mediated apoptosis as additional mode of NK cell killing. 

7.)   The majority of the NK cell killing assays have been performed after 4h co-culture. But figure 1B was performed after 1h. Please explain. 

Author Response

Thank you for your valuable comments. The answer to the question have been uploaded as attached PDF file. For convenience, we also write the full text here.

Point 1: Figure 2C: Please indicate the amount of granzyme B in pg/ml or ng/ml instead of absorbance.

Response 1: We changed the indication of Fig. 2C, from ‘absorbance’ to ‘Granzyme B in ng/ml’.

Point 2: Describe according to table 1 why Hs1bp3 or Cd33 were choicen for comparison. Both proteins are not mentioned in the text at all.

Response 2: Typically, ITIM motifs are often described as (I/V/L/S)xYxx(L/V)[1, 2], but not all ITIMs fit well with this sequences. While murine BST2 satisfies the formula, human BST2 did not show this canonical sequences. To showed that human BST2 is one of the non-canonical ITIM motifs, we referenced other examples of non-canonical ITIM motifs similar to that of human BST2.

Point 3: Explain in the result section why polyI:C was used as treatment for BST2 expression.

Response 3: PolyI:C was used for the purpose of activating NK cells. The increase in BST2 expression was a result of polyI:C treatment, which once again inspired that BST2 could be a potential immune checkpoint molecule.

Point 4: Please introduce the abbreviation BST2 in the abstract and main text.

Response 4: We changed first sentence in abstract from “BST2 is a type II transmembrane protein expressed on various cell types that tethers the release of virus” to “Bone Marrow Stromal Cell Antigen 2 (BST2) is a type II transmembrane protein expressed on various cell types that tethers the release of virus”

And we also changed first sentence in introduction from “BST2 restricts the release of viruses at the plasma membrane” to “Bone Marrow Stromal Cell Antigen 2 (BST2; also known as tetherin, CD317, PDCA-1 and HM1.24) restricts the release of viruses at the plasma membrane”

Point 5: Line 54: Please change “…can recognize a variety of antigens.” To “a variety of ligands” and give some examples of NK ligands.

Response 5: We changed Line 54 from “Activating receptors such as NKG2D or natural cytotoxicity receptors (NCRs) can recognize a variety of antigens” to “Activating receptors such as NKG2D or natural cytotoxicity receptors (NCRs) can recognize a variety of ligands such as MICA for NK2GD[12] and hemagglutinin(HA) of influenza virus for NCR1[13].”

Point 6: Line 55: add death receptor-mediated apoptosis as additional mode of NK cell killing.

Response 6: We changed Line 55 from “NK cells can then directly eliminate target cells by secreting granules such as granzymes and perforin.” to ”NK cells can then directly eliminate target cells by secreting granules such as granzymes and perforin or by death receptor-mediated apoptosis[14].”

Point 7: The majority of the NK cell killing assays have been performed after 4h co-culture. But figure 1B was performed after 1h. Please explain.

Response 7: We stimulated NK cells in two ways; PolyI:C stimulation and IL-2 stimulation. The graph in figure 1B is from NK cells stimulated with polyI:C and the other graphs are from NK cells stimulated with IL-2. NK cells stimulated with polyI:C showed more cytotoxicity to target cells compared to NK cells stimulated with IL-2, so we adjusted the E:T ratio and incubation time to display an appropriate window. In both conditions, Bst2 knockout NK cells showed enhanced cytotoxicity compared to wildtype NK cells.

Reviewer 4 Report

In this study, the authors demonstrate the novel role of BST2 in regulating NK cells’ cytotoxicity. Overall this manuscript is well written and provides novel insights about BST2 as an inhibitory receptor on NK cells. Authors showed that BST2 expression upregulates on activated NK cells and has an ITIM motif in the cytoplasmic tail region. Furthermore, BST2-/- NK cells have shown higher cytotoxicity as compared to their normal counterpart due to higher expression of cytotoxic granules and better degranulation potential. Finally, gene knockout and in-vitro studies demonstrated that the long isoform of BST2 rather than the short isoform act as an inhibitory receptor for NK cells.

The authors need to corroborate these findings using human NK cells. Checking BST2 receptor expression on human NK cells after stimulation and performing cytotoxicity experiments with or without blocking the BST2 receptor would further strengthen this manuscript.  

Author Response

Thank you for your valuable comments. The answer to the question have been uploaded as attached PDF file. For convenience, we also write the full text here.

>Point: The authors need to corroborate these findings using human NK cells. Checking BST2 receptor expression on human NK cells after stimulation and performing cytotoxicity experiments with or without blocking the BST2 receptor would further strengthen this manuscript.

>Response: We sincerely agree with the reviewer’s suggestion. For BST2 to be a good therapeutic target, our findings on murine NK cells should equally apply to human NK cells. The same line of research will be conducted on the human NK cell line, NK92, before trials on human primary NK cells. As the reviewer suggested, performing cytotoxicity experiments on human NK cells may strengthen our results.

Round 2

Reviewer 1 Report

Although the authors responded to my comments in the text, the revised manuscript has not improved in response to any of those my comments raised.

Response 1: Yes, galectin-9 expression varies by cell type. Basically, galectins are kind of secretory molecule. So we detected galectin-9 expression by western blotting method. We found that all target tumor cell lines used in this study express galectin-9, although their expression levels were quite different. Of note, RMA-S also expressed galectin-9 on its surface.

Request 1: Those data need to be shown, as essential information.

Response 2: We plan to elucidate the detailed mechanism of BST2 on NK cell inhibition. We cannot provide a definite answer to this question at this time, but further research directions will be discussed in Response 4. We also analyzed intracellular levels of IFN-gamma in WT and KO NK cells with and without stimulation. We found that intracellular levels of IFN-gamma increased after polyI:C stimulation, but still there was no difference between WT and KO NK cells. Levels of IFN-gamma increased after meeting the target cells (in this case, YAC-1 cells). Although there were some fluctuations, still no differences were found between WT and KO NK cells. Nevertheless, we cannot be sure that BST2 does not affect the production of IFN-gamma.

Request 2: Can you share those preliminary observations by commenting on the manuscript?  It is important to strengthen the preferential involvement of BST2 in NK cell cytotoxicity function.

Response 3: We made two independent monoclones of the galectin-9 knock-out YAC-1. These two clones showed complete depletion of galetin-9 expression and were used as target cells in this study. Since YAC-1 cells were used as a main target cells for NK cell cytotoxicity test throughout this study, we decided to knocked out galectin-9 gene in this cell line to maintain consistency of data analysis.

Request 3: Then, those results must be included (KO efficacy).

Response 4: Thanks for the critical comment. We are currently actively searching for answer(s) as an independent study. One of the ideas we are currently standing on is that most activating receptors on NK cells do not possess their own signaling motifs. Instead, they transmit the signal through common adapter molecules such as DAP-10 or DAP-12 which contain ITAM motifs. These adapter molecules are not restricted to a specific activating receptor. Therefore, controlling these adapter molecules would be a clever way to regulate NK cell cytotoxicity initiated by various types of ligands. When NK cells to a target cell, an immunological synapse occurs at the interface between the two cells. Immune synapses are regions rich in lipid raft, and lipid rafts are where many NK cell receptors are located. DAP-10 and DAP-12 both are found at lipid raft boundaries. It is really interesting that BST2 is also abundant in lipid raft boundaries due to its topology. The N-terminus of BST2 is in non-lipid raft region, whereas the C-terminal GPI anchor of BST2 is embedded in lipid raft. By its function, BST2 is known as an organizer of membrane microdomains. Another idea is the possibility of competition between BST2 and adapter molecules in the recruitment of signaling molecules. DAP-12 has two crucial tyrosine residue in its ITAM motif. With strong signaling, both tyrosine residues are phosphorylated and DAP-12 transmits the activation signal. However, when the signal is weak and only one tyrosine residue is phosphorylated, an inhibitory signal is transmitted. The partial sequence of DAP-12 ITAM motif has SDVYSDL(SxxYxxL), which coincidently resembles with the sequence in ITIM of BST2; SYDYCRV(SxxYxxV). Although we do not have critical experimental data at the moment, many circumstantial evidence inspires that BST2 may affect the action of adaptor molecules. Based on these previous findings, we are currently investigating exactly which molecule(s) interacts with BST2 at the cellular level.

Request 4:  Then, should you revise the manuscript by discussing those points.

Author Response

Thank you for your consideration. We have updated manuscript as requested. The latest version of manuscript is uploaded seperately and changes are listed here.

Request 1: Those data need to be shown, as essential information.

Response 1: We added western blotting images to Figure 3-E (upper panel) and added description to the manuscript.

Page 5-6, Line 199-201: "All three wildtype target tumor cell lines used in this study (RMA-S, B16 and YAC-1) showed expression of Gal-9 (Figure 3-E, upper panel)"

Request 2: Can you share those preliminary observations by commenting on the manuscript?  It is important to strengthen the preferential involvement of BST2 in NK cell cytotoxicity function.

Response 2: We added comment about those observations to the manuscript.

Page 4, Line 137-140: "We also examined the intracellular expression of interferon gamma (IFN-γ) in Bst2+/+ and Bst2-/- NK cells. Although IFN-γ expression increased after stimulation, both showed comparable levels of IFN-γ in the naïve state and after polyI:C stimulation (data not shown). "

Request 3: Then, those results must be included (KO efficacy).

Response 3: We added western blotting images to Figure 3-E (lower panel) and added description to the manuscript.

Page 6, Line 199-201: "and Lgals9-/- YAC-1 showed complete loss of Gal-9 expression upon evaluation by western blotting (Figure 3-E, lower panel)."

Request 4:  Then, should you revise the manuscript by discussing those points.

Response 4: We added comments about those points in the discussion section.

Page 10-11, Line 350-371: Many activating receptors on NK cells do not possess their own signaling motifs. Instead, they transmit the signal through noncovalently associated common adaptor molecules, such as DAP-10 or DAP-12, which contain ITAM motifs [35]. These adaptor molecules are not restricted to a specific activating receptor, therefore, controlling these adaptor molecules would be a clever way to regulate NK cell cytotoxicity initiated by various types of ligands. When NK cells attach to a target cell, an immunological synapse occurs at the interface between two cells [36]. Immune synapses are regions rich in lipid rafts where many NK cell receptors are located [37]. Both DAP-10 and DAP-12 are found at lipid rafts, in particular DAP-12 is reported to be found in its boundaries [38, 39]. It is interesting that BST2 is also abundant in the boundaries of lipid rafts due to its topology [40, 41]. The C-terminal GPI anchor of BST2 is embedded within the lipid raft region, whereas the N-terminus is located outside of lipid raft [40]. With this characteristic, BST2 can perform its function as an organizer of membrane microdomains [42]. DAP-12 has two crucial tyrosine residues in its ITAM motif. With strong signaling, both tyrosine residues are phosphorylated, and DAP-12 transmits the activation signal. However, when the signal is weak, only one tyrosine residue is phosphorylated and inhibitory signal is transmitted [27]. The partial sequence of DAP-12 ITAM motif has SDVYSDL(SxxYxxL), which coincidently resembles with the sequence in ITIM of BST2; SYDYCRV(SxxYxxV). It might be possible that BST2 somehow interfere NK cell adaptor molecules such as DAP-12 by direct or indirect interactions in the lipid raft. Another possibility would be the competition between BST2 and DAP12 for the phosphorylation of signaling motifs, thus resulting in the regulation of NK cell cytotoxicity.

Round 3

Reviewer 1 Report

Most of my requests for improving the manuscript were responded to accordingly.